

# Kinetics of anti-SARS-CoV-2 antibodies and hematological parameters in hospitalized pre-vaccination COVID-19 patients in Peru

Salyoc Tapia-Rojas[1,2,3], Alexis Germán Murillo Carrasco[4], Maria J. Pons[5], Manuel Ugarte-Gil[6,7] and Ana Mayanga-Herrera[1,2]

[1] Cancer and Stem Cells Research Group, Universidad Cientifica del Sur, Lima, Peru
[2] Laboratorio de Cultivo Celular e Inmunologia, Universidad Cientifica del Sur, Lima, Peru
[3] Facultad de Medicina, Universidad San Martin de Porres, Lima, Peru
[4] Centro de Investigação Translacional em Oncologia (LIM24), Departamento de Radiologia, Universidade de São Paulo, Sao Paulo, Brazil
[5] Grupo de Enfermedades Infecciosas Re-Emergentes, Universidad Cientifica del Sur, Lima, Peru
[6] Grupo Peruano de Estudio de Enfermedades Autoinmunes Sistémicas, Universidad Cientifica del Sur, Lima, Peru
[7] Rheumatology Department, Hospital Guillermo Almenara Irigoyen, Lima, Peru

Corresponding authors
Salyoc Tapia-Rojas,
stapiar@cientifica.edu.pe
Ana Mayanga-Herrera,
amayanga@cientifica.edu.pe

## ABSTRACT

**Background**. The COVID-19 pandemic exposed vulnerabilities in health systems and revealed variations in immune responses across populations worldwide. This study examined the kinetics of IgG and IgM antibodies against S1 and receptor-binding domain (RBD) proteins in hospitalized Peruvian patients prior to vaccination.

**Method**. A total of 157 serological samples were collected from 44 hospitalized COVID-19 patients during Peru's first wave (August–October 2020) and stored at −80 °C. Anti-SARS-CoV-2 IgG and IgM antibodies were quantified using an in-house ELISA with recombinant Spike S1 and RBD proteins. Statistical analyses—including linear regression, Kaplan-Meier curves, and receiver operating characteristic (ROC) curves—were conducted to evaluate antibody kinetics, clinical correlations, and predictive accuracy.

**Results**. IgG levels stabilized between days 10 and 15 of hospitalization, while IgM levels declined after day 10, with greater variability observed in severe acute respiratory distress syndrome (ARDS) cases. A significant positive correlation was found between IgG levels and lymphocyte counts ($R = 0.37$, $p < 0.001$), and a negative correlation with neutrophil counts ($R = -0.33$, $p < 0.01$), particularly in severe ARDS non-ICU patients ($R = -0.34$, $p < 0.01$). Severe ARDS cases also exhibited elevated neutrophil-to-lymphocyte ratios and increased inflammatory biomarkers, such as C-reactive protein and D-dimer, indicating an exacerbated inflammatory response associated with poorer prognosis. Risk factors, including sex and obesity, were linked to higher mortality and increased need for mechanical ventilation. This study contributes to a better understanding of the immune response in COVID-19 and supports the development of predictive models based on immunological and hematological biomarkers.

## INTRODUCTION

In December 2019, an outbreak of severe acute respiratory syndrome (SARS) caused by a novel coronavirus, SARS-CoV-2, emerged in Wuhan, China, leading to COVID-19 (*Ray & Bhattacharya, 2024*; *Wu, Chen & Chan, 2020*). This virus quickly escalated into a global public health threat due to its high transmissibility and the severe clinical complications it induces, including respiratory, neurological, cardiovascular, microvascular, and gastrointestinal effects, many of which have been fatal (*Silaghi-Dumitrescu et al., 2023*; *Hua et al., 2021*). By 2023, more than 750 million cases of SARS-CoV-2 infection were reported worldwide (*World Health Organization, 2024*). However, the precise number of deaths remains uncertain, as some studies suggest these figures may be overestimated due to other underlying causes of mortality (*Lewnard, Kang & Laxminarayan, 2023*; *Rossen et al., 2020*). The virus has continued to mutate throughout the pandemic, producing new variants that have sustained its global presence (*Hillary & Ceasar, 2023*). Consequently, numerous scientific and medical organizations have mobilized to improve understanding of the virus and to develop effective diagnostic tools, treatments, and preventive measures, including vaccines (*Pinilla et al., 2021*; *Bloom et al., 2021*).

The immune response is critical in defending the body against viral infections such as SARS-CoV-2. This response has two main components: the innate immune response, which acts rapidly and non-specifically as the first line of defense (*Müller & Schultze, 2023*), and the adaptive immune response, which is more antigen-specific, takes longer to activate, and provides long-term immune memory. Memory B and T lymphocytes persist in circulation after infection, allowing for a faster and more efficient response upon reinfection with the same pathogen. However, the duration of humoral and cellular immune memory varies depending on the microorganism, the severity of the infection and the characteristics of the host (*Abbas, Litchman & Pillai, 2021*). Previous studies have reported an early decline in humoral immunity against SARS-CoV-1 and MERS-CoV, while research on humoral immunity against SARS-CoV-2 is ongoing, particularly concerning genetic variations and population differences (*Müller & Schultze, 2023*; *Maison, Deng & Gerschenson, 2023*; *Merad et al., 2022*).

In Peru, significant research has been conducted on the management, treatment, and investigation of COVID-19 throughout the pandemic (*Pons et al., 2021*; *World Health Organization, 2023*). However, most studies have primarily focused on seroprevalence and hematological parameters during the early stages of the pandemic in Peru (*Huarcaya-Victoria et al., 2023*; *Moyano et al., 2023*; *Huamaní et al., 2024*; *Álvarez Antonio et al., 2021*). To date, no reports have examined the kinetics of antibodies and the kinetics of hematological parameters in the Peruvian population before SARS-CoV-2 vaccination. This information is essential for understanding baseline humoral responses, the temporal profiles of these antibodies, and their correlation with clinical parameters, offering valuable insights into fundamental immune response mechanisms in underrepresented populations.

This study aimed to analyze the kinetics of IgM and IgG antibodies against Spike protein subunit 1 (anti-S1) and the receptor-binding domain (anti-RBD) of SARS-CoV-2, as well as the evolution of hematological parameters in hospitalized COVID-19 patients in Peru.

Additionally, we evaluated the relationship between these factors and disease severity, comparing groups of patients with moderate acute respiratory distress syndrome (ARDS), severe ARDS non-ICU and ICU, all before SARS-CoV-2 vaccination in Peru.

## MATERIALS & METHODS

### Study population and sample

This retrospective observational cohort study included a total of 157 serological samples were collected from 44 hospitalized COVID-19 patients at the Hospital Nacional Guillermo Almenara Irigoyen in Lima, Peru, between August and October 2020, during the first wave of infections and prior to the start of vaccination programs in Peru. The diagnosis was confirmed through antibody immunochromatography tests or qPCR. Samples were properly coded, labeled, and stored at −80 °C in the Cell Culture and Immunology Laboratory at the Universidad Científica del Sur in Lima, Peru, *Pons et al. (2021)*.

Patients were classified as having critical disease according to the Clinical Management of COVID-19 guidelines by the World Health Organization (WHO). This classification includes individuals with acute respiratory distress syndrome (ARDS). Based on the degree of oxygenation impairment and ICU admission status, patients were further stratified into three groups: 10 moderate ARDS, 24 severe ARDS non-ICU, and 10 severe ARDS ICU. Moderate ARDS was defined by a $PaO_2/FiO_2$ ratio between 100 mmHg and 200 mmHg, while severe ARDS was characterized by a $PaO_2/FiO_2$ ratio of $\leq$100 mmHg (*World Health Organization, 2023*). These criteria were applied uniformly to ensure a standardized classification across patient groups.

### Clinical and hematological data

Data were gathered from clinical records, including age, sex, hospitalization date, progression date to disease severity, and discharge or death date. Additionally, comorbidities such as hypertension, coronary disease, chronic respiratory disease, diabetes, and obesity were documented. hematological parameters were also collected, including levels of leukocytes, lymphocytes, hemoglobin, neutrophils, platelets, D-dimer, prothrombin time (PT), activated partial thromboplastin time (aPTT), C-reactive protein (CRP), ferritin, creatinine, neutrophil-to-lymphocyte ratio (NLR) and oxygen saturation to fraction of inspired oxygen ratio ($SpO_2/FiO_2$). The clinical-epidemiological characteristics, cytokine levels, and commercial anti-Spike antibody ratios of this cohort have been previously reported (*Pons et al., 2021*) at baseline. In the present study, we performed additional analyses not reported previously, including the assessment of antibody kinetics using in-house ELISA assays, temporal trajectories of hematological markers, correlation matrix construction, principal component analysis (PCA), survival analysis, receiver operating characteristic (ROC) curve analysis, and the development of a predictive nomogram.

### IgG and IgM antibody levels by enzyme-linked immunosorbent assay

To analyze IgG and IgM antibody levels, serological samples stored at −80 °C from the 44 hospitalized patients were used. An in-house enzyme-linked immunosorbent assay

(ELISA) kit was developed using recombinant SARS-CoV-2 Spike S1 and RBD proteins to detect anti-S1 and anti-RBD IgG and IgM antibodies. Following a protocol adapted from *Stadlbauer et al. (2020)*, 96-well plates were coated with 50 µL of recombinant protein (one µg/mL) and incubated overnight at 4 °C, followed by PBS-Tween washing. Plates were then blocked with PBS-Tween-20 containing 5% skim milk for 2 h. Heat-inactivated serum samples (56 °C for 30 min, diluted 1:101 in PBS-Tween) were added to the plates for a 2-h incubation. Serum samples were heat-inactivated at 56 °C for 30 min, diluted 1:101 in PBS-Tween, and added to the plates for a 2-hour incubation. After washing, goat anti-human IgG or IgM conjugated with horseradish peroxidase (HRP) was added as the secondary antibody and incubated for 1 h. The plates were then washed again, the o-phenylenediamine dihydrochloride (OPD) substrate was added for a 10-minute dark incubation. The reaction was stopped with HCl, and absorbance was measured at 490 nm. This protocol was utilized to develop kits for detecting IgG and IgM antibodies against S1 and RBD proteins. The in-house ELISA assays were validated using a panel of negative sera ($n = 56$) and clinically confirmed SARS-CoV-2-positive samples ($n = 161$). Results were compared with those of an FDA-approved commercial anti-S1 IgG ELISA kit (2606-9601G; EUROIMMUN, Lübeck, Germany), demonstrating strong correlation. Intra-assay variability was assessed by testing replicates of the same samples within a single plate, while inter-assay variability was determined across plates on different days.

## Statistical analysis

Microsoft Excel version 16.84 was used for data collection and organization, while RStudio version 4.3.3 facilitated statistical analysis. Descriptive statistics were presented in tables, and inferential statistics included both bivariate and multivariate analyses. Data distribution was initially assessed using Kolmogorov–Smirnov normality tests. Linear regression analyses were then performed to examine IgG and IgM antibody trends over the first 35 days of hospitalization, with mean values used to detect significant changes over time. Box plots compared IgM and IgG levels and ratios (IgM/IgG) among patient groups (moderate ARDS, severe ARDS ICU and non-ICU) over time, with group differences analyzed using the Kruskal-Wallis test (for multiple groups) or the Mann–Whitney test (for two-group comparisons), with significance set at $p < 0.05$.

Dispersion analyses with trend lines were conducted to explore correlations between various variables, utilizing Pearson or Spearman correlation coefficients based on data distribution. A heatmap illustrated correlations among multiple variables using Pearson coefficients. Kaplan–Meier curves were generated to assess survival probabilities and mechanical ventilation needs based on clinical and hematological parameters, with log-rank tests identifying significant group differences ($p < 0.05$). Principal component analysis (PCA) was performed to visualize patient distributions across clinical variables. A nomogram was developed to predict clinical events based on antibody levels and clinical parameters, supporting individualized interpretation. Finally, receiver operating characteristics (ROC) curves assessed prediction model accuracy at 4, 7, and 10 days of hospitalization, with high areas under the curve (AUC) indicating strong predictive performance.

The sample size was determined based on the availability of hospitalized COVID-19 patients during the study period. Missing data were addressed through multiple imputation when appropriate. Cases with extensive missing values were excluded from specific analyses. Since this was a retrospective cohort study, there was no loss to follow-up; however, missing data due to incomplete medical records were accounted for as described above.

## Bias and limitations

To minimize selection bias, all eligible hospitalized patients from the study period were included without preselection. Information bias was controlled by using standardized medical records and validated serological assays for antibody relative quantification. However, due to the retrospective nature of the study, missing data could not be entirely avoided, and confounding factors, such as individual treatment regimens, were not fully controlled.

## Ethical aspects

This research study was approved by the Institutional Review Board at Universidad Científica del Sur (No. 343-CIEI-CIENTÍFICA-2020) and Universidad San Martín de Porres (0582-2024-CIEI-FMH-USMP). Due to the health emergency, the requirement for informed consent was waived by the COVID-19 Institutional Review Board at IETSI-EsSalud (42-IETSI-ESSALUD-2020).

# RESULTS

## Clinical characteristics and comorbidities

A total of 157 samples from 44 symptomatic COVID-19 patients hospitalized at Guillermo Almenara Irigoyen National Hospital (HNGAI) were analyzed. Patients with critical disease were divided into three groups according to their severity and ICU admission: Moderate ARDS ($n = 10$), Severe ARDS non-ICU ($n = 24$), and Severe ARDS ICU ($n = 10$). Table 1 details clinical characteristics, pre-existing conditions, treatments, and outcomes. Median age and gender distribution were similar across groups, and 91% received corticosteroids without significant differences between groups. Mortality was higher in ICU patients (90%) compared to others ($p < 0.001$). Obesity was significantly more frequent in ICU patients ($p = 0.043$), with no differences in diabetes, hypertension, or other comorbidities. Biomarker values, including hematological, inflammatory, and respiratory parameters, were assessed longitudinally, with medians used for central tendency due to variable measurement frequencies and data distribution.

## Kinetics of anti-SARS-CoV-2 antibodies and hematological parameter evolution in hospitalized patients by disease severity

The in-house ELISA for IgG anti-S1 showed a sensitivity of 92.82% (95% CI [88.03–96.12]) and specificity of 95.65% (95% CI [85.16–99.47]). For IgG anti-RBD, sensitivity was 91.16% (95% CI [86.04–94.86]) and specificity 91.30% (95% CI [79.21–97.58]). The IgM anti-S1 assay had lower sensitivity at 47.20% (95% CI [39.30–55.22]) but high specificity at 96.43% (95% CI [87.69–99.56]). IgM anti-RBD showed a sensitivity of 61.03% (95% CI [53.51–69.04]) and specificity of 94.64% (95% CI [85.13–98.88]). The intra-assay and

**Table 1  Clinical characteristics and comorbidities of patients hospitalized with COVID-19 at HNGAI.**

| Variable | Overall $N = 44^a$ | Moderate ARDS $N = 10^a$ | Severe ARDS non-ICU $N = 24^a$ | Severe ARDS ICU $N = 10^a$ | p-value[b] |
|---|---|---|---|---|---|
| Age | 56 (44, 64) | 60 (31, 69) | 56 (45, 63) | 55 (41, 65) | >0.9 |
| Gender | | | | | 0.8 |
| Female | 5/44 (11%) | 2/10 (20%) | 2/24 (8.3%) | 1/10 (10%) | |
| Male | 39/44 (89%) | 8/10 (80%) | 22/24 (92%) | 9/10 (90%) | |
| Corticosteroids treatment | 40/44 (91%) | 8/10 (80%) | 22/24 (92%) | 10/10 (100%) | 0.5 |
| Mechanical ventilation | 10/44 (23%) | 0/10 (0%) | 0/24 (0%) | 10/10 (100%) | <0.001 |
| Deceased | 11/44 (25%) | 0/10 (0%) | 2/24 (8.3%) | 9/10 (90%) | <0.001 |
| Obesity | 15/44 (34%) | 2/10 (20%) | 6/24 (25%) | 7/10 (70%) | 0.043 |
| Diabetes mellitus | 3/44 (6.8%) | 1/10 (10%) | 1/24 (4.2%) | 1/10 (10%) | 0.6 |
| High blood pressure | 11/44 (25%) | 2/10 (20%) | 5/24 (21%) | 4/10 (40%) | 0.5 |
| Chronic kidney disease | 1/44 (2.3%) | 0/10 (0%) | 1/24 (4.2%) | 0/10 (0%) | >0.9 |
| Lung disease | 1/44 (2.3%) | 0/10 (0%) | 0/24 (0%) | 1/10 (10%) | 0.5 |
| Cardiovascular disease | 1/44 (2.3%) | 0/10 (0%) | 0/24 (0%) | 1/10 (10%) | 0.5 |
| Gout | 1/44 (2.3%) | 0/10 (0%) | 1/24 (4.2%) | 0/10 (0%) | >0.9 |
| Other | 3/44 (6.8%) | 1/10 (10%) | 1/24 (4.2%) | 1/10 (10%) | 0.6 |

Notes.
[a] Median (Q1, Q3); n/N (%).
[b] Kruskal–Wallis rank sum test; Fisher's exact test.

inter-assay coefficients of variation were lower than 5% and 10%, respectively, across all analytes. The IgG anti-S1 assay correlated strongly with the commercial ELISA kit (Spearman rho = 0.85). These results support the reliability of the in-house assay for detecting SARS-CoV-2-specific antibodies, particularly IgG responses.

Levels of IgG against RBD and S1 progressively increased in all patient groups, reaching a plateau around 10 to 15 days of hospitalization. Patients with severe ARDS disease, particularly those admitted to the ICU, exhibited notably higher IgG levels compared to moderate ARDS and severe ARDS non-ICU patients, with significant differences between groups at various time points ($p < 0.05$). In contrast, IgM showed an early rise followed by a progressive decline from day 10 onward, with a faster reduction in moderate ARDS and severe ARDS non-ICU patients (Figs. 1 and 2).

Regarding the evolution of hematological parameters varied with disease severity, leukocytes, and the neutrophil-to-lymphocyte ratio (NLR) showed an early increase in severe ARDS patients, with persistently elevated values over time in ICU patients ($p < 0.001$). This pattern suggests a heightened immune and stress response in severe ARDS cases. In contrast, lymphocyte counts significantly declined in the early stages of hospitalization across all groups, followed by gradual recovery, particularly in moderate ARDS cases, which may indicate a lower inflammatory burden in these patients ($p < 0.05$). Inflammatory markers, including C-reactive protein (CRP), ferritin, and D-dimer, reached elevated peaks in the early days of hospitalization in severe ARDS cases, especially in ICU patients, reflecting an intense inflammatory and procoagulant response. Significant

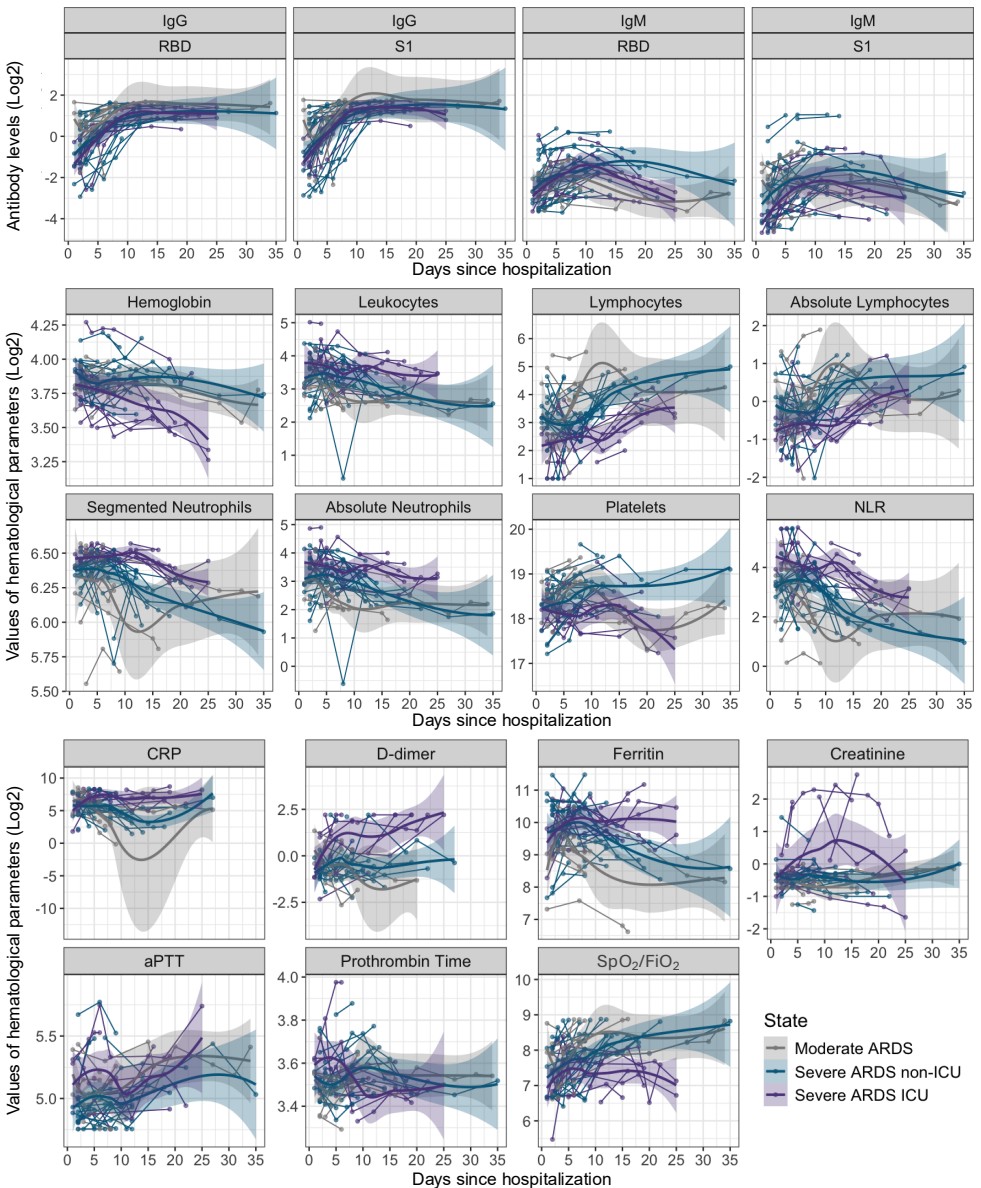

**Figure 1** **Kinetics of antibody levels and hematological parameters evolution in patients hospitalized with COVID-19 over 35 days.** The first panel shows the trend lines for IgG and IgM antibody levels (anti-RBD and anti-S1), and the others panels present the trend lines for hematological parameter values, both displayed on a log2 scale. Data are stratified by disease severity: moderate ARDS (gray), severe ARDS non-ICU (blue), and severe ARDS ICU (purple). Shaded areas around the trend lines represent 95% confidence intervals.

differences were observed between severity groups, particularly for CRP and ferritin during the first 10 days ($p < 0.001$), indicating a more aggressive activation of inflammation and coagulation pathways (Fig. 1) (Table S1).

Creatinine levels increased in severe ARDS patients, particularly in those in the ICU, suggesting renal dysfunction associated with disease severity. Although the differences

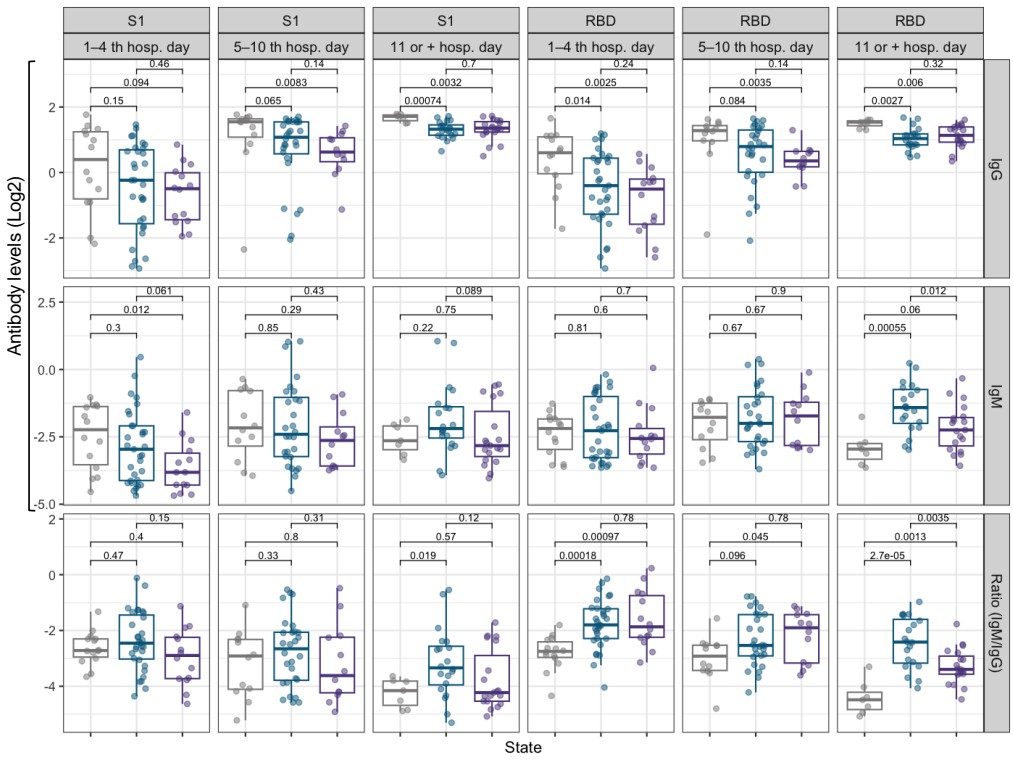

**Figure 2** **Box plots comparing IgM and IgG antibody levels, as well as the IgM/IgG ratio for the RBD and S1 subunits, across three hospitalization intervals in COVID-19 patients.** < 5 days (baseline), 5–10 days, and > 10 days. Data are stratified by disease severity and ICU admission status: moderate ARDS (gray), severe ARDS non-ICU (blue), and severe ARDS ICU (purple).

did not exhibit statistical significance at certain points during hospitalization periods, an upward trend was evident, which could indicate a higher risk of renal failure in critical cases. Prothrombin time (PT) and activated partial thromboplastin time (aPTT) did not show significant changes between groups, suggesting that coagulation alterations are primarily related to inflammatory activation rather than intrinsic or extrinsic coagulation pathways. The $SpO_2/FiO_2$ ratio remained consistently lower in severe ARDS patients, especially those in the ICU, throughout the observation period, reflecting persistent respiratory dysfunction in these individuals ($p < 0.05$). This finding underscores the severity of pulmonary impairment in critical patients and the need for continuous respiratory monitoring (Fig. 1).

We further analyzed the IgM/IgG ratio for the RBD and S1 domains to assess the transition from the acute immune response, marked by IgM production, to a more stable, long-term IgG response (Fig. 2). In the early days of hospitalization, severe ARDS ICU and non-ICU patients exhibited a higher IgM/IgG ratio. However, after day 5, this ratio significantly decreased across all groups ($p < 0.05$), reflecting a rise in IgG and a decline in IgM. This trend was most pronounced in moderate ARDS patients, whose ratio reached

the lowest values after 10 days, suggesting a faster resolution of the acute phase. In contrast, severe ARDS ICU patients maintained a higher ratio for longer, indicating extended immune activation.

## Correlation analysis of antibody levels and clinical parameters in hospitalized COVID-19 patients

Correlation analysis revealed significant associations between IgM and IgG antibody levels (anti-S1 and anti-RBD) and key clinical parameters in hospitalized COVID-19 patients, stratified by disease severity (Fig. 3A).

Notably, an inverse correlation was observed between anti-S1 and anti-RBD IgG levels and the NLR ($R = -0.39$ and $R = -0.40$, respectively; $p < 0.001$), across all groups, particularly in severe ARDS cases. Additionally, IgG anti-S1 levels were inversely correlated with segmented neutrophil counts ($R = -0.33$, $p < 0.001$) and positively correlated with lymphocyte counts ($R = 0.37$, $p < 0.001$), suggesting an adaptive immune response influenced by disease severity (Fig. 3B, top panels).

Furthermore, a positive correlation between IgG anti-S1 levels and the $SpO_2/FiO_2$ ratio was observed in moderate ARDS ($R = 0.41$, $p < 0.05$) and severe ARDS non-ICU ($R = 0.53$, $p < 0.01$) patients, which indicates a link between antibody response and respiratory function (Fig. 3B, middle left panel); however, no such correlation was found in severe ARDS UCI patients. Similarly, IgM anti-RBD levels positively correlated with platelet counts ($R = 0.37$, $p < 0.001$), especially in moderate ARDS and severe ARDS ICU patients, suggesting potential associations with coagulation and inflammation pathways (Fig. 3B, middle right panel).

Additionally, CRP levels displayed a positive association with D-dimer levels ($R = 0.42$, $p < 0.001$), underscoring the interplay between systemic inflammation and coagulation activation across patient groups (Fig. 3B, bottom left panel). NLR also demonstrated robust correlations with CRP ($R = 0.52$, $p < 0.001$) and ferritin ($R = 0.49$, $p < 0.001$), particularly in moderate ARDS and severe ARDS non-ICU patients (Fig. 3B, bottom right panel), highlighting the relationship between systemic inflammation and hematological changes. Lastly, the $SpO_2/FiO_2$ ratio exhibited a significant inverse correlation with NLR ($R = -0.39$, $p < 0.001$), particularly in severe ARDS non-ICU patients, suggesting that elevated inflammatory responses may coincide with worsened respiratory function.

## Principal component analysis of immunological and clinical parameters in hospitalized COVID-19 patients

Principal component analysis (PCA) revealed that the first two dimensions captured 41.5% of the total variance (26.8% for Dim1 and 14.7% for Dim2). In the two-dimensional space defined by these components, there was notable overlap among moderate ARDS, severe ARDS non-ICU, and severe ARDS ICU patient groups, suggesting limited differentiation based on these parameters alone (Fig. 4).

However, certain trends emerged, indicating that patients classified as Severe ARDS ICU tended to cluster at more positive values along Dim1, potentially reflecting heightened inflammatory burden and immune response. Key contributors to these dimensions included lymphocytes, anti-RBD IgG, segmented neutrophils, absolute neutrophil count,

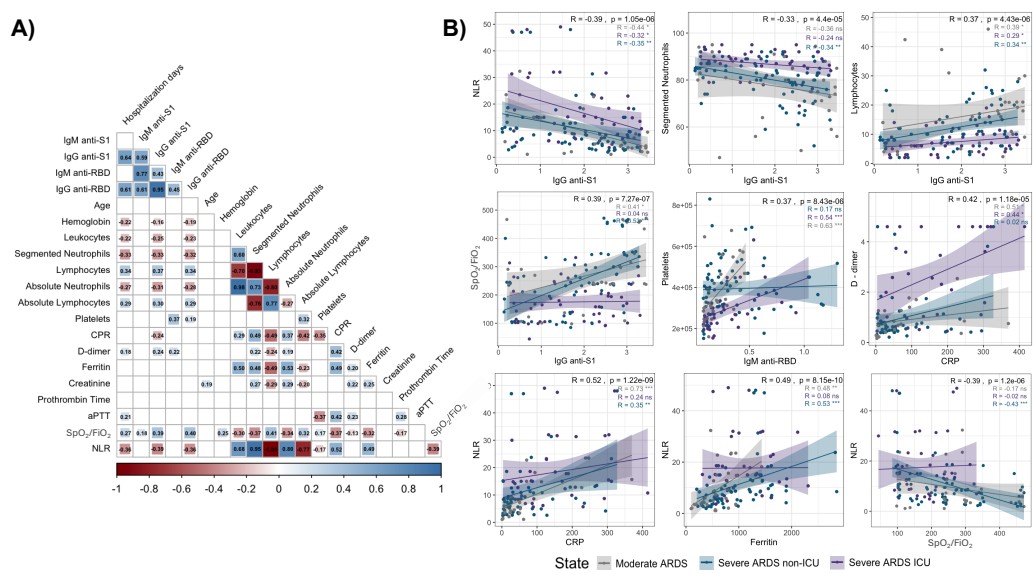

**Figure 3  Correlation matrix and scatter plots of antibody levels and clinical parameters in hospitalized COVID-19 patients.** The heatmap (A) shows correlation coefficients (R values) between IgM and IgG antibodies (anti-S1 and anti-RBD) and various clinical parameters. Positive correlations are indicated in blue, and negative correlations in red, with the intensity reflecting the strength of the association. Non-significant correlations are represented as blank spaces. The scatter plots (B) highlight significant relationships, with regression lines and color-coded disease severity: moderate ARDS (gray), severe ARDS non-ICU (blue), and severe ARDS ICU (purple). Correlation coefficients (R) and *p*-values: $p < 0.05$ (*), $p < 0.01$ (**), and $p < 0.001$ (***).

anti-S1 IgG, and the NLR, underscoring the role of immune and inflammatory responses in group separation.

Inflammatory markers such as CRP and D-dimer, along with immune response indicators (anti-RBD IgG and IgM), exhibited particularly strong associations with severe ARDS cases, aligning with established clinical patterns of COVID-19. These findings suggest that while certain biomarkers partially differentiate severity, the immune and inflammatory responses in COVID-19 remain complex, as evidenced by the overlap among clinical groups.

## Survival and mechanical ventilation probability based on clinical and hematological parameters in COVID-19 patients

Kaplan–Meier curves reveal that advanced age ($\geq 56$ years), low hemoglobin levels (<14.1 g/dL), obesity, elevated CRP ($\geq 67$ mg/mL), low lymphocyte count (<9.3), high ferritin ($\geq 877$ µg/L), high absolute neutrophil counts ($\geq 8.9$), and elevated D-dimer levels ($\geq 0.75$ µg/mL) significantly impact survival and mechanical ventilation requirements in hospitalized COVID-19 patients (Fig. 5).

Patients aged 56 years or older (Fig. 5A; $p = 0.0086$), those with low hemoglobin (Fig. 5B; $p = 0.0052$), obesity (Fig. 5C; $p < 0.0001$), or elevated CRP levels (Fig. 5D; $p < 0.0001$) had significantly reduced survival and a higher likelihood of requiring ventilation. Also, low lymphocyte counts (Fig. 5E; $p = 0.031$), high ferritin (Fig. 5F;

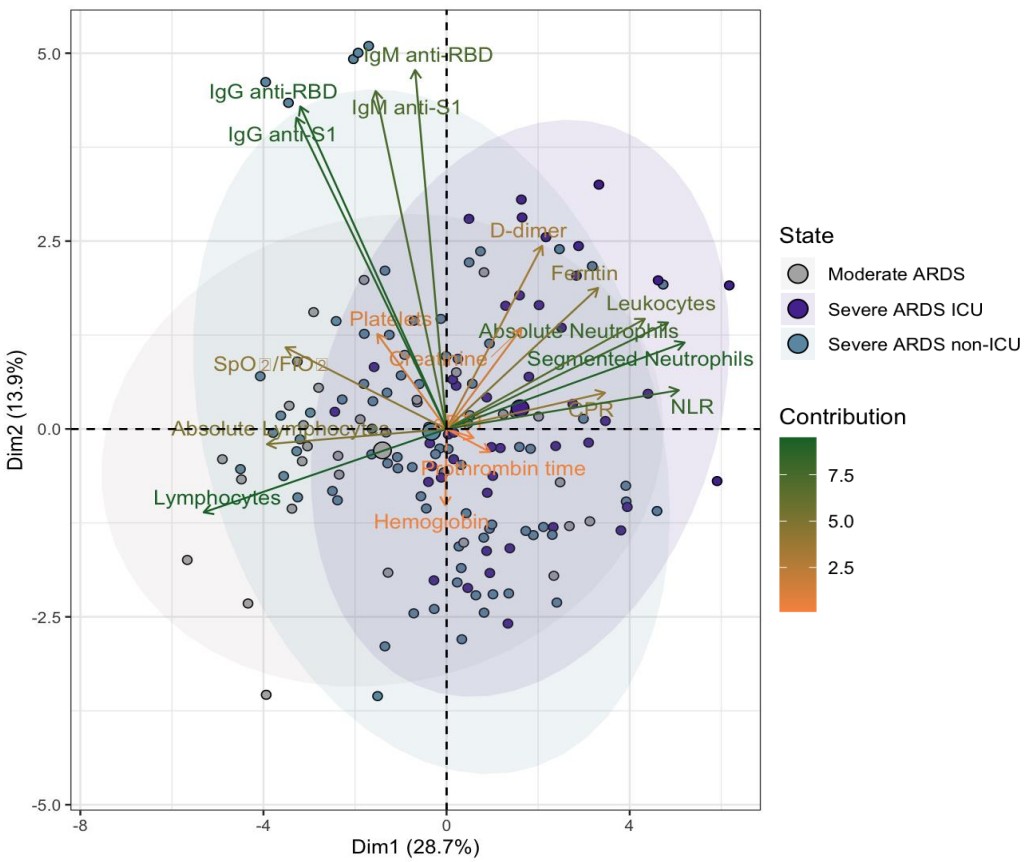

**Figure 4  Principal component analysis (PCA) of parameters in hospitalized COVID-19 patients.** The first principal component (Dim1) accounts for 28.7% of the total variance, and the second component (Dim2) explains 13.9%. Patients are grouped into three categories: moderate ARDS (gray), severe ARDS non-ICU (blue), and severe ARDS ICU (purple). Arrows represent variable contributions, with dark green indicating high contributors and orange for lower contributors. Ellipses denote 95% confidence intervals for each group, illustrating the spread and overlap in the PCA space.

$p = 0.0016$), elevated neutrophil counts (Fig. 5G; $p = 0.00056$), and high D-dimer levels (Fig. 5H; $p = 0.00024$) were each linked to poorer survival rates and increased probability of ventilation, underscoring the impact of inflammation and coagulation abnormalities in severe ARDS COVID-19 cases. These findings highlight the prognostic value of these clinical and hematological markers, which could facilitate early risk stratification and help identify patients more likely to experience severe ARDS outcomes.

## Nomogram for predicting adverse outcomes in hospitalized COVID-19 patients

The developed nomogram incorporates IgG anti-S1 levels, which indicate humoral immune response to SARS-CoV-2; obesity status (0 or 1, with 1 denoting obesity); ferritin, where high levels may suggest systemic inflammation or acute infection; and the NLR, an indicator of immune response dynamics. Each of these variables contributes a specific point value to

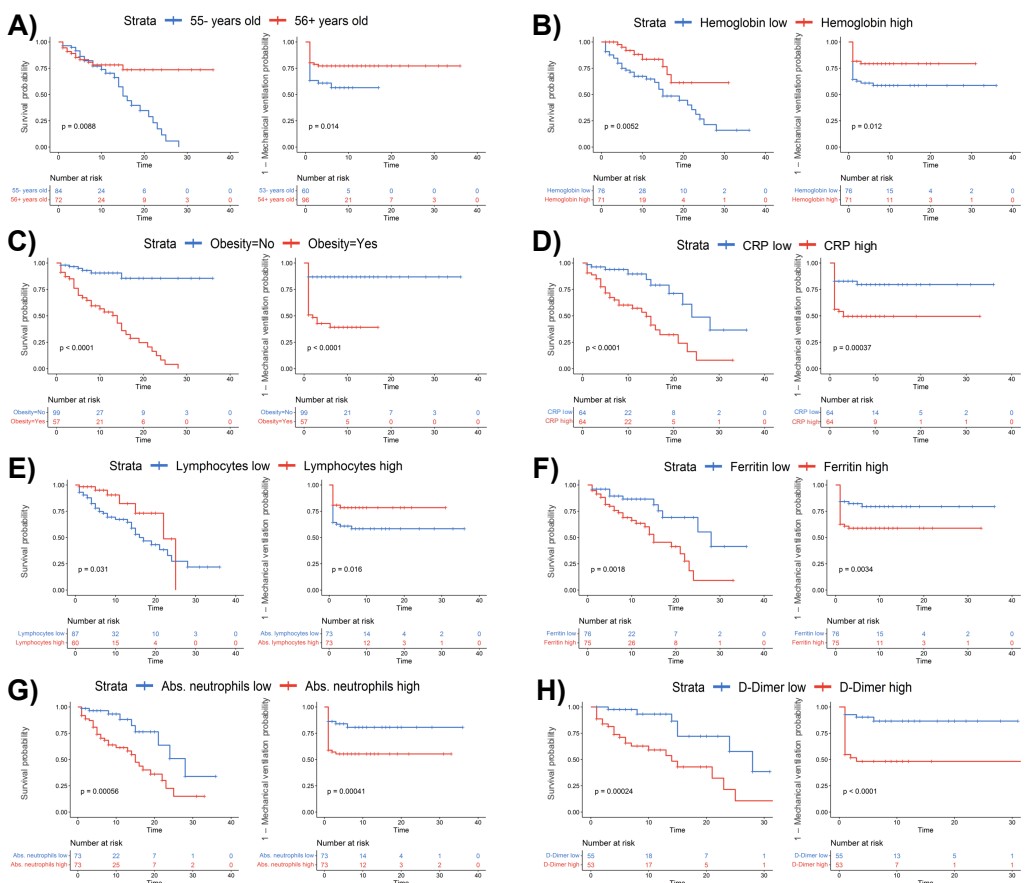

**Figure 5  Kaplan–Meier survival analysis stratified by clinical and laboratory parameters in COVID-19 patients.** Survival curves are shown for patients stratified by (A) age (≤ 55 *vs.* >56 years), (B) hemoglobin levels (low *vs.* high), (C) obesity status (no *vs.* yes), (D) C-reactive protein (CRP) levels (low *vs.* high), (E) lymphocyte counts (low *vs.* high), (F) ferritin levels (low *vs.* high), (G) absolute neutrophil counts (low *vs.* high), and (H) D-dimer levels (low *vs.* high). Blue and red lines represent patients with lower and higher values, respectively. Log-rank *p*-values are provided in each panel. The number of patients at risk at each time point is shown below each plot.

a cumulative patient score, which is then translated into a linear predictor to estimate the probability of adverse outcomes over hospitalization.

For illustrative purposes, a patient presenting with obesity (assigned a score of 1; ∼80 points), elevated ferritin levels (*e.g.*, 2,800 ng/mL; ∼30 points), a high neutrophil-to-lymphocyte ratio (*e.g.*, 30; ∼60 points), and increased anti-S1 IgG titers (*e.g.*, 2.5; ∼10 points) would yield a cumulative nomogram score of approximately 180 points. According to the model, this score corresponds to an estimated 90% probability of in-hospital mortality. As shown in Fig. 6, this example illustrates the clinical utility of the nomogram in synthesizing immunological and hematological markers into a quantitative tool for individualized risk stratification and early intervention during COVID-19 hospitalization.

ROC curves were analyzed at 4-, 7-, and 10-days post-hospitalization to assess the model's predictive accuracy, producing AUC values of 0.97, 0.95, and 0.86, respectively.

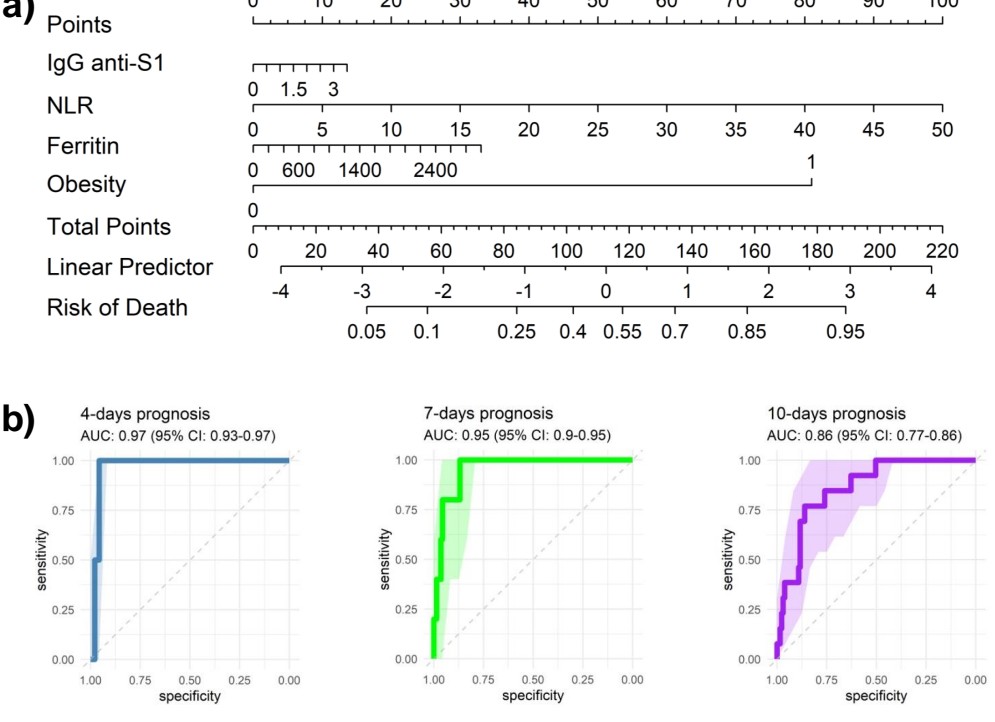

**Figure 6** **Predictive modeling of clinical outcomes.** (A) Nomogram illustrating the predicted probability of clinical events based on key clinical and hematological parameters. (B) Receiver operating characteristic (ROC) curves evaluating the predictive performance of models at 4, 7, and 10 days, showing the area under the curve (AUC) with 95% confidence intervals (95% CI).

These consistently high AUC values highlight strong early predictive accuracy, although a slight reduction in AUC over time may indicate a decrease in predictive power as patients' clinical trajectories evolve.

## DISCUSSION

The pandemic caused by SARS-CoV-2 has triggered an unprecedented global health crisis, severely affecting global health systems. This crisis has underscored humanity's vulnerability to emerging infectious diseases and the variability in the immune response between different populations worldwide (*Wu, Chen & Chan, 2020*; *Hopman & Mehtar, 2020*).

In this study, IgG antibody levels—specifically anti-S1 and anti-RBD—showed a consistent increase across all groups of hospitalized COVID-19 patients, reaching a plateau between days 10 and 15, suggesting a robust adaptive immune response to SARS-CoV-2. This finding is consistent with other studies that have shown a stabilization of IgG levels during the acute phase of infection (*Garcia et al., 2022*; *Ravichandran et al., 2021*; *Favresse et al., 2021*; *Wang et al., 2020*). The stability of IgG levels could indicate that the humoral immune response is sufficient to neutralize the virus and could prevent disease progression (*Wajnberg et al., 2020*; *Isho et al., 2020*).

On the other hand, IgM levels (both anti-S1 and anti-RBD) exhibited greater variability, generally peaking around day 10 and then declining, consistent with IgM kinetics observed in other populations (*Wang et al., 2020*; *Seow et al., 2020*). This response was particularly variable in severe ARDS ICU cases, which may reflect immune dysregulation in these patients (*Liu et al., 2020*). The fluctuating IgM levels in fatal cases suggest that an ineffective initial immune response might correlate with worse clinical outcomes (*Ravichandran et al., 2021*).

Notably, significant differences in antibody levels were observed between patients with moderate ARDS and severe ARDS, including both ICU and non-ICU. Moderate ARDS patients exhibited higher IgG levels, possibly reflecting a more robust adaptive immune response. While some studies have reported elevated antibody levels in severe cases, likely due to increased antigenic stimulation (*Seow et al., 2020*; *Ripperger et al., 2020*), others have noted a decline in antibody levels among critically ill patients (*De Greef et al., 2021*). These differences could be attributed to variations in antibody measurement methods or the timing of sample collection (*Ward et al., 2022*). Additionally, it is important to consider that most patients in our study received corticosteroid treatment early during hospitalization, which may have influenced antibody responses, particularly in severe ARDS cases (*De Greef et al., 2021*; *Patel et al., 2021*); however, in contrast to previously thought, recent evidence suggests that corticosteroids do not have a long-term impact on antibody levels (*Long et al., 2020*), although they do affect inflammatory biomarkers (*Nakamoto et al., 2024*).

The IgM/IgG ratio also provided insight into immune response kinetics. A decline in this ratio from day 5, more rapid in moderate ARDS cases, suggests that moderate ARDS patients experience faster resolution of the acute infection phase. In contrast, severe ARDS patients, particularly those in ICU, showed a prolonged response, possibly due to higher viral loads and systemic inflammation, exacerbating disease severity (*Liu et al., 2020*; *Asif et al., 2023*; *Longchamp et al., 2020*).

The differences in antibody levels between ICU and non-ICU severe ARDS groups became more evident in the later stages of hospitalization, especially after day 10. These findings highlight the importance of monitoring IgG and IgM levels throughout disease progression, as they may serve as valuable indicators for identifying patients at greater risk of severe ARDS COVID-19. The development of personalized treatment strategies based on the evolution of these antibody levels could significantly improve the prognosis of patients (*Abril et al., 2024*).

The analysis of the correlation between IgM and IgG antibody levels and various clinical and hematological parameters in COVID-19 patients provides valuable insights into the interaction between humoral immunity and disease progression. A moderate positive correlation was observed between IgM and IgG isotypes, notably between IgM anti-S1 and IgG anti-S1, as well as IgM RBD and IgG RBD. These findings are consistent with the expected immune response to viral infections, where both IgM and IgG increase simultaneously in the early infection stages, indicating coordinated immune activation (*Maison, Deng & Gerschenson, 2023*; *Primorac et al., 2022*). The simultaneous increase in

IgM and IgG suggests a transition from an acute response, primarily mediated by IgM, to a more stable, long-lasting phase dominated by IgG (*Qin et al., 2021*).

The strong correlation between IgG anti-S1 and IgG anti-RBD levels suggests that both antibodies target critical viral regions, reflecting a coordinated response against the most immunogenic SARS-CoV-2 epitopes (*Freeman et al., 2021*; *Iyer et al., 2020*). This pattern suggests a robust and sustained immune response, as documented in previous studies of antibody kinetics in viral infections (*Favresse et al., 2021*; *Wang et al., 2020*; *Wajnberg et al., 2020*; *Isho et al., 2020*).

Regarding clinical parameters, a significant positive correlation was found between the duration of hospitalization and IgG anti-S1 and IgG anti-RBD levels. This suggests that patients who survive longer hospital stays tend to exhibit a more sustained humoral response. However, this does not necessarily indicate a protective role, as disease severity itself could influence IgG production. In severe ARDS ICU patients, a delayed or dysregulated immune response may result in prolonged inflammation, potentially explaining this association (*Asif et al., 2023*; *Çetin et al., 2021*; *Rees et al., 2020*). Additionally, the positive correlation between IgG anti-S1 and lymphocyte count highlights that elevated IgG anti-S1 levels may reflect effective adaptive immune activation, marked by increased lymphocyte counts. B lymphocytes, which produce antibodies, are known to proliferate during infection, resulting in higher concentrations of specific immunoglobulins (*Zhao et al., 2016*; *Lapuente, Winkler & Tenbusch, 2024*). Conversely, the negative correlation between IgG anti-S1 and segmented neutrophils may indicate that an effective humoral response is accompanied by reduced neutrophil-mediated inflammation, a pattern previously linked to milder disease severity (*Laing et al., 2020*; *Chen et al., 2020b*). This modulation of neutrophil activity in the later stages of the immune response aligns with improved clinical prognosis (*Chen et al., 2020b*).

A moderate positive correlation was also observed between IgM anti-RBD levels and platelet counts. Reactive thrombocytosis, commonly seen in viral infections such as COVID-19, may reflect the role of platelets in both innate and adaptive immunity (*Manne et al., 2020*). Elevated platelet counts could thus indicate systemic inflammation, which aligns with elevated IgM levels in the early stages of infection (*Chen et al., 2020b*; *Barrett et al., 2021*). However, the moderate strength of this correlation suggests it may be part of a more complex immune response rather than a dominant pattern.

For lung function parameters, a positive correlation was noted between the $SpO_2/FiO_2$ ratio and IgG S1 and IgG RBD levels, suggesting that a rapid, effective immune response is linked to better respiratory function in COVID-19 patients (*Chen et al., 2020b*; *Park et al., 2024*). Other studies have similarly shown that efficient immune responses can contribute to favorable lung function recovery (*Park et al., 2024*).

Correlations with variables like age, hemoglobin, leukocytes, ferritin, C-reactive protein, and D-dimer were weak, suggesting that while these factors are important for overall prognosis, they do not significantly influence antibody kinetics in this cohort (*Goshua et al., 2020*). Among severe ARDS ICU patients, high mortality (90%) and obesity prevalence (70%) underscore the need to identify patients with risk factors for adverse outcomes early on, as obesity is known to exacerbate systemic inflammation and respiratory dysfunction

(*Arulanandam, Beladi & Chakrabarti, 2023*). These findings support previous studies indicating that obesity not only intensifies the inflammatory response but also complicates clinical management, increasing mortality in critically ill patients (*Popkin et al., 2020*).

For inflammatory biomarkers, the elevated levels of CRP, D-dimer, and ferritin in ICU patients support the 'cytokine storm' theory, describing an excessive inflammatory response that can lead to acute respiratory distress syndrome (ARDS) and multi-organ failure (*Ravichandran et al., 2021*; *Abril et al., 2024*; *Mehta et al., 2020*). Elevated CRP has been identified as a key marker in severe COVID-19 progression, while high D-dimer levels are often associated with thromboembolic events, a frequent complication among severe cases (*Abril et al., 2024*; *Laing et al., 2020*).

The NLR also demonstrated an early increase in severe ARDS COVID-19 patients, with persistently high levels among those admitted to the ICU, indicating a stronger immune and stress response in these severe ARDS patients. As a marker of inflammation and immune imbalance, NLR correlated with other severity biomarkers, such as D-dimer, ferritin, CRP, and $SpO_2/FiO_2$, which signify systemic inflammation and respiratory dysfunction (*Chen et al., 2020b*; *Yang et al., 2020*). This correlation suggests that NLR, alongside these biomarkers, may be valuable for assessing disease severity and prognosis.

The low lymphocyte counts in ICU patients are especially notable. Lymphopenia, frequently observed in these patients, has been established as a poor prognostic marker in COVID-19, correlating with higher mortality rates. This lymphocyte depletion may result from excessive immune activation in severe ARDS patients (*Wu et al., 2020*; *Chen et al., 2020a*).

Similarly, low $SpO_2/FiO_2$ ratios and hemoglobin levels in severe ARDS patients suggest significant impairments in gas exchange and oxygenation capacity, predicting a higher risk of respiratory failure and the need for mechanical ventilation (*Abril et al., 2024*; *Goshua et al., 2020*). Anemia and hypoxia aggravate the clinical presentation and have been documented as poor prognostic indicators in COVID-19 (*Cavezzi, Troiani & Corrao, 2020*).

Kaplan–Meier survival and mechanical ventilation (MV) analysis revealed that elevated neutrophils, D-dimer, ferritin, and CRP levels are associated with lower survival probabilities and higher MV requirements. These biomarkers are widely recognized indicators of an uncontrolled inflammatory response, which can lead to tissue damage and multi-organ dysfunction (*Sukrisman & Sinto, 2021*; *Paixão et al., 2023*). Conversely, patients with elevated hemoglobin and lymphocyte levels had increased survival probabilities and lower MV rates, suggesting an enhanced ability to sustain an effective immune response against SARS-CoV-2 (*Chen et al., 2020b*). However, it is important to note that the relatively small sample size may have limited the statistical power of these subgroup analyses, potentially introducing variability and reducing the robustness of the observed associations. Thus, these findings should be interpreted with caution and validated in larger, independent cohorts.

Patients under 54 years old had a higher likelihood of requiring mechanical ventilation compared to older patients, which contrasts with studies associating advanced age with poorer COVID-19 outcomes (*Molani et al., 2022*; *Statsenko et al., 2022*). This discrepancy

may be due to study-specific factors like comorbidities (*e.g.*, diabetes, stroke, lung disease), which could increase risks of metabolic and cardiovascular complications, while lung diseases further compromise respiratory function (*Zhou et al., 2020*; *Huang et al., 2021*).

Male sex was identified as a significant risk factor, with men exhibiting lower survival rates compared to women. This disparity may be explained by biological and hormonal differences that influence immune responses and disease progression (*Jun et al., 2021*; *Peckham et al., 2020*). Notably, 88.6% of the hospitalized patients in this study were male, a factor that may have influenced the observed outcomes. Although this male predominance (88.6%) limits the ability to perform robust sex-based comparisons, it is consistent with previous epidemiological reports indicating higher severity and hospitalization rates among male COVID-19 patients. These differences have been linked to variations in immune response modulation and the elevated expression of angiotensin-converting enzyme 2 (ACE2), the functional receptor for SARS-CoV-2, in men. Thus, the sex distribution observed in our cohort likely mirrors the clinical reality of severe disease during the initial wave of the pandemic in Peru, rather than reflecting a selection bias.

In the unsupervised analysis through PCA, we observed an overlap among patients with moderate ARDS, severe ARDS non-ICU, and severe ARDS ICU. This overlap suggests that, despite group-level trends in inflammatory and immunological markers, individual-level variability remains high. Such a pattern reinforces the multifactorial nature of COVID-19 progression, where overlapping immune responses and clinical characteristics complicate clear categorization. Therefore, we emphasize the importance of using integrated multivariate models (such as our proposed nomogram) to capture this complexity and improve predictive performance. The PCA findings also highlight the limitations of unsupervised dimensionality reduction when applied to heterogeneous clinical data in complex diseases such as COVID-19.

ROC curve analysis demonstrated high predictive accuracy of the models for days 4, 7, and 10, affirming their robustness in forecasting the clinical trajectory of patients. These results incorporate multivariate and predictive analyses within a clinical context, offering a comprehensive perspective on the clinical progression of COVID-19 patients (*Xu et al., 2023*; *Cai et al., 2021*).

Overall, the results indicate that the nomogram-based predictive model shows reasonable accuracy in estimating the likelihood of adverse events in COVID-19 patients, with slightly higher accuracy in the short term (4 and 7 days) compared to 10 days. By integrating clinical, immunological, and hematological variables, this model provides a comprehensive approach to risk assessment, which could support clinical decision-making and facilitate personalized patient management (*Xu et al., 2023*; *Tang et al., 2021*).

While this nomogram is valuable for short-term risk stratification in COVID-19 patients, external validation with similar cohorts is essential for broader applicability (*Li et al., 2021*). While the model performed well within this cohort, confirming its robustness across different populations will be crucial to ensuring that the selected variables remain predictive in diverse patient groups.

This study presents certain limitations that should be considered when interpreting the findings. Its retrospective design may introduce selection bias, despite including all eligible

hospitalized patients. Antibody kinetics were analyzed based on days of hospitalization. However, variation in the interval between symptom onset and admission may influence interpretations. Symptom onset dates were not consistently available, limiting our ability to align immune kinetics with disease timeline precisely. Additionally, the limited sample size and absence of a pre-pandemic control group may restrict the generalizability of findings. Missing data from medical records could have influenced some hematological parameters, although standardized assays and hospital records helped mitigate information bias. Although cross-reactivity testing with other respiratory viruses was not performed, the use of recombinant S1 and RBD proteins as coating antigens, and the high specificity observed in comparison with a commercial FDA-approved ELISA, mitigate this limitation. As the study was conducted in a single hospital, its applicability to other populations remains uncertain. Nevertheless, these findings provide valuable insights into antibodies and hematological parameters kinetics and their clinical associations in hospitalized COVID-19 patients before widespread vaccination in Peru. Monitoring antibody levels and inflammatory biomarkers could improve risk stratification and patient management, aiding preparedness for the emergence of new viral variants.

## CONCLUSIONS

In conclusion, this study emphasizes the importance of antibody kinetics and inflammatory biomarkers in understanding COVID-19 progression, showing that IgG stability may protect against severe ARDS outcomes, while fluctuations in IgM levels correlate with immune dysregulation in severe ARDS cases. Monitoring these immune markers, along with clinical and demographic factors, can enhance patient stratification and guide personalized treatment strategies, potentially improving prognosis for severe COVID-19 cases.

### Funding
This study was funded by the "Semilla COVID-19" grant given by Universidad Científica del Sur and "Kaelin Prize 2020" from EsSalud. The funders had no role in study design, data collection and analysis, decision to publish, or preparation of the manuscript.

### Grant Disclosures
The following grant information was disclosed by the authors:
The "Semilla COVID-19" grant given by Universidad Científica del Sur.
"Kaelin Prize 2020" from EsSalud.

### Competing Interests
The authors declare there are no competing interests.

## Author Contributions

- Salyoc Tapia-Rojas conceived and designed the experiments, performed the experiments, analyzed the data, prepared figures and/or tables, authored or reviewed drafts of the article, and approved the final draft.
- Alexis Germán Murillo Carrasco analyzed the data, prepared figures and/or tables, authored or reviewed drafts of the article, and approved the final draft.
- Maria J. Pons conceived and designed the experiments, performed the experiments, authored or reviewed drafts of the article, and approved the final draft.
- Manuel Ugarte-Gil conceived and designed the experiments, authored or reviewed drafts of the article, and approved the final draft.
- Ana Mayanga-Herrera conceived and designed the experiments, performed the experiments, analyzed the data, authored or reviewed drafts of the article, and approved the final draft.

## Human Ethics

The following information was supplied relating to ethical approvals (*i.e.*, approving body and any reference numbers):

Institutional Ethics Committee for Research at Universidad Científica del Sur and Research Ethics Committee for COVID-19 of IETSI-EsSalud.

## Data Availability

The raw measurements are available in the File S2.

## Supplemental Information

Supplemental information for this article can be found online at http://dx.doi.org/10.7717/peerj.19771#supplemental-information.

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
