# Peer review of "Kinetics of anti-SARS-CoV-2 antibodies and hematological parameters in hospitalized pre-vaccination COVID-19 patients in Peru"

_PeerJ, doi:10.7717/peerj.19771_

## Round 0.1 · original submission · Major Revisions

Please respond especially to Reviewer 2's comments regarding the timeliness of this work.

Reviewer 1 ·

Basic reporting

1.1 Clarity & language:
The manuscript is generally well-written and clearly understandable.
1.2 Introduction & context:
The introduction provides sufficient background, explaining the global impact of COVID-19, the basic immune response to SARS-CoV-2, and the importance of studying antibody and haematological kinetics in the Peruvian population who are naïve to COVID-19 vaccination. The authors clearly identify the knowledge gap and the rationale for this study.
1.3 Literature referencing:
The cited literature is generally relevant and appropriately supports the context and rationale for the study.
1.4 Structure:
The manuscript follows the IMRaD format and aligns well for an original article type.
1.5 Figures & tables:
Figure captions are descriptive, and the figures are of appropriate quality. Table 1 and Supplementary Table 1 present key findings clearly and effectively.
1.6 Ethical approval:
The ethical statement is clearly described and well-documented. Waiver of informed consent was appropriate given the emergency context, and supporting documentation was provided.
1.7 Raw data:
The submission includes appropriately summarised datasets, and the analyses appear transparent.

Experimental design

2.1 Research question:
The study addresses a clearly defined and relevant research question: to evaluate the kinetics of anti-SARS-CoV-2 IgM/IgG (anti-S1/RBD) antibodies and haematological parameters in hospitalised, unvaccinated Peruvian COVID-19 patients and assess their correlation with disease severity. This provides valuable data from the Peruvian population during the pandemic's early wave.
2.2 Scope:
The study fits well within the scope of biomedical sciences.
2.3 Rigor (Technical & Ethical):
-The retrospective cohort design is appropriate given the pandemic context but carries inherent limitations (see Section 3).
-Stratification into three severity categories using WHO ARDS criteria and ICU status is appropriate.
-The use of stored samples and hospital records is standard for retrospective studies.
-The in-house ELISA method for antibody quantification is acceptable, though more detail on its validation is needed (see 4.6).
-The statistical methods are generally appropriate for the types of comparisons made.
-Ethical approval and the waiver of consent are acceptable and well documented.
2.4 Methodological transparency:
The methods section provides sufficient detail for replication in terms of patient selection, sample handling, and statistical analysis. However, validation of the in-house ELISA requires additional information (see 4.6).

Validity of the findings

3.1 Data Robustness & Controls:
The dataset includes 157 samples from 44 patients, which strengthens the kinetic assessment.
However, subgroup sample sizes (n=10, 24, 10) limit the statistical power of subgroup comparisons and survival analyses. This limitation is acknowledged by the authors. Severity groups are used as internal comparators, which is a valid approach.
3.2 Statistical soundness:
The use of Kruskal-Wallis, Mann-Whitney, correlation analyses, PCA, Kaplan-Meier curves, and ROC curves is appropriate and well justified.
3.3 Acknowledgment of limitations:
The authors clearly discuss the study’s limitations, including its retrospective and single-centre design, small sample size, missing data, and the inability to adjust for confounders. The marked male predominance (89%) should be emphasised further as it limits the generalisability and analysis of sex-related outcomes.
3.4 Conclusions:
The conclusions are supported by the data and are consistent with prior literature. Findings such as the plateau of IgG by days 10–15, IgM decline after day 10, and associations of inflammatory biomarkers with severity are valid. The discussion appropriately places the results in the context of other populations while highlighting region-specific insights.

Additional comments

4.1 Clarify data reuse:
Please clarify which parts of the dataset have been previously reported (reference #15) and which analyses in the current manuscript are novel.
4.2 Institutional details:
To improve the geographic context, consider explicitly mentioning the city locations of the Hospital Nacional Guillermo Almenara Irigoyen and Universidad Científica del Sur.
4.3 Table 1:
Consider stratifying patient age groups (<60 and ≥60) to provide more information.
4.4 Smoking status:
If smoking status was collected (current, former, never), suggest including this in the table, as smoking has been identified as a potential risk factor for disease severity.
4.5 Impact of small sample size:
While the small sample size is acknowledged, a more explicit statement about its impact on statistical stability, especially for subgroup correlations and Kaplan-Meier analyses, is recommended.
4.6 In-house ELISA validation:
Additional information is needed regarding the performance of the in-house ELISA assay. Please provide validation metrics such as sensitivity, specificity, inter/intra-assay variability, cross-reactivity (e.g., other coronaviruses, influenza or flu-like liiness pathogens), and comparison to a commercial assay, validated assay or gold standard (e.g., neutralisation test).
4.7 Confounding factors:
Given the near-universal use of corticosteroids, a brief discussion on how this may have influenced immune kinetics and inflammatory biomarkers would be helpful. The strong male predominance should also be more prominently noted as a potential confounder.
4.8 Timing of kinetics:
The study uses hospitalisation day as a time reference. Please discuss the potential variation between hospital admission and symptom onset and how this might affect the interpretation of kinetic data.
4.9 Nomogram interpretation:
Consider including a brief example or clearer explanation of how the nomogram could be applied in clinical settings to enhance practical relevance.
4.10 PCA interpretation:
Please discuss the observed overlap between severity groups in PCA plots and its implications for biomarker-based discrimination or prediction.
4.11 Subscript notation:
It is recommended to use subscript formatting (e.g., PaO₂, FiO₂, SpO₂) throughout the manuscript, figure 3 and supplementary table S1.

Reviewer 2 ·

Basic reporting

The kinetics of antibodies against SARS-CoV-2 have been extensively investigated, so this work does not contribute anything new. it is also known that severe ARDS cases displayed an
elevated neutrophil-lymphocyte ratio and increased inflammatory biomarkers, such as C-reactive protein and D-dimer, reflecting an exacerbated inflammatory response associated with poorer prognosis.
Perhaps if this work had been published earlier, in 2021, it might have contributed more to the understanding of the pathophysiology of COVID-19, but not in 2025. I don't understand why it took the authors so long to try to publish their work. The initial SARS-CoV-2 variants were more pathogenic than the current Omicron variants, so this paper is way out of date. Publication in 2025 is less timely, as the overall focus of the research has been on post-vaccination immunity, variants and long COVID.

Experimental design

The experimental design is appropriate

Validity of the findings

The findings of this research are important, but as I mentioned, they should have been published in 2021. In 2025, they are irrelevant as recent Omicron variants are less pathogenic than those that dominated in 2020.

Additional comments

None

---

## Round 0.2 · accepted · Accept

All recommendations were addressed, and the process was completed successfully.

Reviewer 1 ·

Basic reporting

Thank you for thoroughly addressing the concerns I raised in your previous submission. The changes made have improved the clarity and overall quality of the manuscript.

Experimental design

-

Validity of the findings

All concerns have already been clarified.

Additional comments

-